# Impact of Daycare Service Interruption during COVID-19 Pandemic on Physical and Mental Functions and Nutrition in Older People with Dementia

**DOI:** 10.3390/healthcare10091744

**Published:** 2022-09-11

**Authors:** Ya-Shin Wang, Cheng-Fu Lin, Fu-Hsuan Kuo, Ying-Chyi Chou, Shih-Yi Lin

**Affiliations:** 1Center for Geriatrics & Gerontology, Taichung Veterans General Hospital, Taichung 40705, Taiwan; 2Executive Master of Business Administration, Tunghai University, Taichung 40704, Taiwan; 3Department of Business Administration, Center for Healing Environment Administration and Research, Tunghai University, Taichung 40704, Taiwan; 4Institute of Clinical Medicine, School of Medicine, National Yang Ming Chiao Tung University, Taipei 11221, Taiwan

**Keywords:** COVID-19, daycare service, functional decline, cognitive impairment, nutritional status

## Abstract

This study evaluated changes of cognitive, physical, and nutritional status before and after the interruption and resumption of daycare services during the COVID-19 pandemic in older dementia people in a daycare center. Comprehensive geriatric assessment data were analyzed before and after the lockdown of daycare center services, including mini-mental state examination, activities of daily living (ADL) scores, mini-nutritional assessment-short forms (MNA-SF), and timed up-and-go (TUG) tests. Among 19 dementia people participating in daycare services, 17 participants were enrolled in the study with, finally, two excluded because of incomplete follow-ups. They had a median age of 81 years; their MNA-SF scores and TUG values deteriorated significantly after a 3-month closure of daycare services (*p* < 0.05), and after resumption of daycare services the MNA-SF scores and TUG values recovered to near the pre-lockdown levels (*p* < 0.05). Besides, baseline ADL scores predicted a decline and recovery of TUG and MNA-SF values. Our findings suggest that planning continuous support for older dementia adults is important for daycare facilities during COVID-19 pandemic confinement.

## 1. Background

Dementia is a global issue; there were over 50 million people with dementia worldwide in 2020 [1]. Dementia can cause disabilities in memory, attention, judgement, language, and abstract thought. Furthermore, it increases disability-adjusted life years and has a substantial impact on personal living quality and care needs [2]. The Global Burden of Disease report had revealed a high burden of disability caused by dementia, which was greater than almost any other conditions [3,4]. However, recent studies showed that through appropriate interventions, such as improving healthcare delivery and evidence-based cost-effective resource allocation, the age-standardized mortality and disability rates can be decreased in people with dementia [5,6]. In Taiwan, the population of people with dementia had reached 280,000 in 2018; moreover, the population of people with dementia will double to exceed 460,000 in 2031 [7]. The characteristics of dementia are deterioration of cognitive function and the subsequent decline in physical activities, which are not only a national burden but also a family care problem [8]. In response to the phenomenon, the government launched long-term care (LTC) plan version 1.0 in 2008 and upgraded to LTC plan version 2.0 in 2016 [9,10]. The main purposes of the LTC plan version 2.0 was to provide a home- and community-based service (HCBS) system, adopting fee-for-service payment, which could allow many older adults to age in place by preventing isolation, depression, and undue cognitive and physical decline among older community-dwelling adults [10]. A three-layer HCBS service network was set to provide the 17 types of services within towns and districts [11]. Among the various LTC service models currently in use, daycare services (DCS) are designed to meet the daily living and social needs of adults with functional limitations during the day with a professionally supported environment. Several studies have shown that DCS allows disabled elderly people (e.g., dementia patients) to maintain close contact with their home environment in the community. Moreover, DCS can reduce the time and burden of caregiving, and increase caregivers’ life satisfaction [12,13,14]. In a recent study, we showed that multiple non-pharmacological activities of DCS over a 6-month period were associated with short-term maintenance of physical and mental functions in older people with dementia or disability in a daycare center. Furthermore, the family caregivers’ burden was reduced [9].

The novel coronavirus disease 2019 (COVID-19) was discovered in late 2019 and resulted in a pandemic [15]. Importantly, the COVID-19 pandemic not only affected medical care systems worldwide, but also had a hugely disruptive impact on daily life and negatively affected psychological health [16,17,18]. In older people, home quarantine that attempted to prevent severe COVID-19 illness may also simultaneously place these individuals at a greater risk of social isolation. Particularly, older persons in nursing homes or those receiving LTC services had to face such challenges with considerable deleterious effects on the mental and physical health of this vulnerable group [18,19]. Therefore, it was crucial to collect relevant data using instruments such as the comprehensive geriatric assessment (CGA) to evaluate any changes in older people to assess whether withdrawal of care services had any long-term consequences. Physical inactivity, poor sleep quality, and low psychological wellbeing have been associated with adverse outcomes because of social isolation in older age [20,21,22]. Recent cross-sectional studies on the consequences of lockdown measures have relied on participants’ recollections of their prior states to infer how lockdown measures may have induced changes. However, objective evidence of serial changes in physical and psychosocial behavioral variables before and after restriction and resumption of LTC services is currently less studied [3,23].

On 21 January 2020, Taiwan reported its first confirmed case of COVID-19, which prompted the Taiwan Centers for Disease Control to establish the Central Epidemic Command Center with inter-departmental horizontal coordination involving the ministries of the interior, education, transportation and communications, etc. [24]. After May 2021, new cases of COVID-19 infection increased dramatically to 15,674 on July 31, according to the Central Epidemic Command Center. To effectively mitigate the effects of COVID-19, thorough border management, health inspections, contact tracing, and other public health measures were announced with restricted indoor/outdoor gatherings for eating, recreation, meetings, and learning at school. Accordingly, LTC services such as DCS were affected. We hypothesized that it is possible the restrictions during the COVID-19 pandemic may exacerbate psychological reactions and physical dysfunction in disabled elderly people. In the present study, serial comprehensive geriatric assessments (CGA) were performed and aimed at measuring longitudinal changes in physical and cognitive and nutritional parameters before and after restriction and resumption of daycare services in older people with dementia during the COVID-19 pandemic.

## 2. Methods

### 2.1. Study Design

This was a prospective cohort study performed at a hospital-affiliated daycare center in central Taiwan between 1 April 2021 and 31 December 2021. The flow diagram is shown in Figure 1. During the COVID-19 pandemic, the CGA response and differences were analyzed by comparing the CGA of participants attending day care before the outbreak of COVID-19 in April in Taiwan, with CGA measured about 3 months after the interruption of daycare services. The CGA was done within one week when the older individuals came back to our daycare center once a peak in cases had subsided in August. Moreover, the CGA scores were also evaluated in the third month after reopening, i.e., in December. All individuals were informed of all aspects of the study and they provided informed consent. The study was approved by the Institutional Review Board of the medical center (IRB no: CG20244B).

### 2.2. Recruitment

Inclusion criteria were individuals who were aged 55 years or older with mild to moderate disability or dementia and who participated in daycare services in our hospital. The level of disability was rated on a scale of 2 to 8 according to LTC version 2.0 in Taiwan [11]. The severity of dementia was determined by the clinical dementia rating (CDR) [25]. Exclusion criteria were incomplete or interrupted follow-ups leading to missing data at the end of the study.

### 2.3. Setting

The regular daycare center investigated in this study was situated near a hospital with its own staff and allowed a maximum number of 20 participants who could receive DCS. In addition, the daycare center had access to outdoor areas, with a patio and garden. The daycare services of the daycare center were multicomponent and designed by the staff working there. Every program was approximately one hour in duration and was provided twice per week from Monday to Friday. Programs included reminiscence therapy, exercise therapy, cognitive occupational therapy, horticultural therapy, art therapy, and music therapy [9].

### 2.4. Measures

At recruitment, the demographic information of the participants, including age, gender, body height, body weight, body mass index (BMI), marital status, residential status, and assistive equipment, were collected. Before the suspension of daycare services because of the COVID-19 outbreak, CGA was routinely done in April to serve as the baseline data. When the daycare center was reopened in August, the same CGA was performed by the same well-trained staff at the 1-week and 3-month follow-ups. In brief, the items in the CGA measured cognition, mood, functional capacity, nutrition, and health status. Cognition was measured by the mini-mental state examination (MMSE), with scores ranging from 0 to 30, and consisted of the following categories: orientation, registration, attention/calculation, recall, language, repetition, and complex commands [26,27]. The MMSE score of less than or equal to 24 indicated cognitive impairment if the participant was literate or less than or equal to 13 if illiterate [26]. Mood was evaluated by the five-item geriatric depression scale (GDS-5), which consisted of five questions, and depressive symptoms were defined as a GDS-5 score greater or equal to 2 [28,29]. Functional capacity was assessed by the Barthel index (BI) of activities of daily living (ADL) and the Lawton instrumental ADL (IADL) scale. The range of BI was 0 to 100 and included 10 common activities: feeding, bathing, grooming, dressing, bowel control, bladder control, toileting, chair transfer, ambulation, and stair climbing [30]. The Lawton IADL scale was 0 to 8, with lower scores indicating poorer ability and composed of eight areas: meal preparation, ordinary housework, managing finances, managing medications, phone use, stairs, shopping, and transportation [31,32]. Lower extremity function mobility was evaluated by the 6 m walking-speed test (6MWS), the functional reach test, and the timed up-and-go (TUG) test to measure both static and dynamic balance [31]. A value of the 6MWS less than 1 m/s indicated low physical performance [33]. The FRT was one of the most commonly used to assess balance and more than 15 cm might predict a fall [34,35]. The TUG test required the participants to rise from a chair, walk straight for 3 m, return back to the chair, and finally then sit. If they walked longer than 30 s to complete the test, it meant the participants could not go outside alone and required a gait aid [36]. Nutritional status was assessed by the mini-nutritional assessment-short form (MNA-SF), which contained six questions regarding a decrease in food intake, weight loss, mobility, psychological distress or acute disease, neuropsychological problems, and body mass index [37]. The scores ranged from 0 to 14 and scores less than 12 indicated a risk of malnutrition [9]. Life quality was measured by the EQ-5D instruments, which were developed by the EuroQol Group and had two parts. One was the three-level version of the EQ-5D (EQ-5D-3L), which comprised five dimensions: mobility, self care, usual activities, pain/discomfort, and anxiety/depression; this was then was converted to a single utility index ranging from 0 to 1. The other one was the visual analogue scale (VAS), a self-perception scale from 0 to 100, with zero representing the worst health and 100 representing the best health [9,38]. The validity and reliability of the Chinese versions of MMSE, GDS-5, ADL, IADL, MNA-SF, and EQ-5D-3L all had been reported in the literature [27,29,30,37,38].

### 2.5. Statistical Analysis

Continuous variables are expressed as the mean and the standard error of the mean (mean ± SEM), median, and interquartile range (IQR, 25–75%). Categorical data are expressed as the number and percentage of the total number of participants. Paired comparisons were made using the Friedman test for continuous variables and the Dunn–Bonferroni test as a post hoc test. The relationships between the various baseline parameters of CGAs and the participants’ physical function, cognitive function, and nutritional status 3 months after resumption of DCS were analyzed by Spearman’s correlation analysis. Statistical analyses were performed using SPSS version 22.0 (SPSS Inc., Chicago, IL, USA). The statistical significance was set at *p* < 0.05.

## 3. Results

During the recruitment period, 19 participants were reviewed for eligibility and 2 participants were excluded as they did not complete the study. Finally, a total of 17 participants were enrolled in the study.

### 3.1. Description of the Study Participants

The demographic profiles of the participants attending daycare services are shown in Table 1. The participants’ median age was 81 years (IQR: 74.5–86 years), and 58.8% were women. The majority (88.2%) were literate and 15 (29.4%) had graduated from senior high school. The marital status of participants was as follows: 35.3% married, 58.8% widowed, and 5.9% divorced. The most common comorbidities were hypertension, diabetes mellitus, and Parkinson’s disease. The median LTC disability level was 4 (IQR: 4–5) and the median CDR was 1.0 (IQR: 0.5–1.5).

### 3.2. Differences in Physical and Cognitive Functions, and Nutritional Status at Baseline, during Restriction and after Resumption of Daycare Services in the COVID-19 Outbreak

Table 2 shows the CGA at different time points: baseline, after resumption of day care service within 1 week, and 3 months later. Overall, there were significant differences in MMSE, ADL, MNA-SF scores, and TUG values (*p* < 0.05). The Bonferroni post hoc test identified a minor decline of MMSE and ADL scores (one point each) during the restrictions of the COVID-19 pandemic; however, MNA-SF (decreased by 8.8%) and TUG scores (decreased by 6 s) were the worst during the 3-month restriction. Three months after resumption of daycare services, MNA-SF and TUG were all significantly improved in comparison with the values measured during the interruption period and before the closure of daycare services.

### 3.3. Relevant Factors of Physical and Cognitive Function after Restriction and Resumption of Daycare Services

Using univariable analysis, it was found that baseline physical functions, including ADL (r_s_ = −0.61, *p* = 0.009), IADL (r_s_ = −0.54, *p* = 0.025), 6MWS (r_s_ = −0.84, *p* < 0.001), and EQ-5D values (r_s_ = −0.60, *p* = 0.011; r_s_ = −0.74, *p* = 0.001), were related to the decline of TUG after the interruption of daycare services (Table 3). Moreover, it was found that baseline physical functions, including ADL (r_s_ = −0.66, *p* = 0.004), IADL (r_s_ = −0.75, *p* = 0.001), 6MWS (r_s_ = −0.70, *p* = 0.002), and EQ-5D values (r_s_ = −0.55, *p* = 0.023; r_s_ = −0.63, *p* = 0.007), were also related to the recovery of TUG, and BMI (r_s_ = 0.73, *p* = 0.001) was related to the recovery of MNA-SF three months after resumption of daycare services (Table 3).

## 4. Discussion

This observational prospective study provided evidence of longitudinal changes of physical and mental functions, as well as nutritional status in older people with dementia following the suspension of daycare services during the COVID-19 pandemic. In this study, we found that there was a significant decline of TUG and MNA-SF with a minor decline of MMSE and ADL scores after three months’ suspension of daycare services. Three months after resumption of daycare services, TUG and MNA-SF nearly recovered to the baseline values. Moreover, it was shown that the baseline physical function, as measured by the ADL/IADL scores, and the 6MWS value predicted the decline and recovery of the TUG and MNA-SF values. This observation indicates that physical function and nutritional status might be more vulnerable in elderly people with DCS. Several studies, including ours, have shown that DCS is beneficial for older people with dementia and/or disability [9,12,13,14]. Importantly, nutritional status plays a particularly important role in future disability and functional decline [39,40,41]. However, due to the COVID-19 pandemic and the suspension of DCS, older people faced several difficulties, such as performing their regular activities and the inconvenience of obtaining food, which led to a decline in physical and mental function [42,43].

Between the interruption and resumption of DCS, there were significant changes in physical function, as measured by TUG, while ADL scores were marginally decreased. However, the baseline ADL status predicted the TUG decline. In this study, the average duration of daycare service interruption was about 3 months, prior to which there were several regular multidisciplinary programs that were implemented. It is possible that physical parameters, such as TUG, may be more vulnerable to interruption, but for basic activities of living this duration may not be long enough to significantly affect ADL scores [44]. A recent study observed that individuals who spent more time doing sedentary activities had lower step counts and heart rates during the COVID-19 lockdown [45]. Furthermore, even this short-term reduction in activity results in rapid muscle mass loss and physical decline, even in younger adults [46]. Importantly, after resumption of daycare services with multidimensional intervention programs, physical function recovered to the previous level before the interruption of daycare programs. A previous follow-up result in a population of older Japanese showed that many of these older individuals showed notable resilience during the COVID-19 pandemic [47], but older adults who were living alone and were socially inactive were not resilient, and they may be at high risk for incident disability. This information may help prioritize subgroups for health promotion or for permission to go outside in the event that home confinement is required in the future (i.e., the oldest elderly, those whose housing condition is poor, or individuals with diabetes). Overall, when there is a long interruption of rehabilitation programs, as happened in those conducted in daycare centers, healthcare staff should pay close attention to patients’ physical functions.

In line with previous studies on older adults who are institutionalized or living at home, our patients exhibited significant worsening of nutritional status after discontinuation of daycare services [48,49]. Malnutrition among the elderly is a vicious cycle, even in individuals not suffering directly from COVID-19, as it is associated with weight loss and fragility, which in turn worsen a person’s health status. The mechanisms of malnutrition may be multifactorial, including psychosocial factors, oral disorders, polypharmacy, and other associated comorbidities. Therefore, it is essential to provide early promotion of measures to treat malnutrition to prevent the development of more severe conditions. In our patients, there were marginal changes in cognitive function measured by MMSE between pre- and post-lockdown. However, a recent study with a longer observation period that investigated residents living in a long-term care institution showed progressive decline of cognition functions, particularly in those with moderate dementia and severe dementia [50]. The average duration of daycare service interruption in this study was about 3 months. We speculate that this period of interruption may have been too short to cause a significant decline in MMSE scores. In addition, it is possible that some of the patients made use of telephone and video consultations during the discontinuation period, which may have made them more aware of the importance of preserving their health status in a home- and outpatient-care setting rather than in the context of a nursing institution [51]. Although the observed deterioration in MMSE scores may be the result of a natural process associated with the progression of the underlying disease, it should be noted that cognitive and physical functions still decreased relatively quickly and significantly, which may be due to isolation, lack of social contact, and less regular physical activity. There is clearly a need to minimize the adverse effects of social isolation, especially in older people with dementia, who would likely benefit from assistance to mitigate the risk of worsening cognitive outcomes.

A significant change of at least 4 points for MMSE [52,53], 4.9 s for TUG [54], 7.8% change for MNA-SF [55], and 16.2 points for ADL [56,57] between two measurements has been reported in the literature to indicate actual deterioration of the patients. With these criteria, in our study, TUG and MNA-SF declined significantly by 6 s and 8.8%, respectively, after three months’ suspension of daycare services, while MMSE, and ADL scores only showed minor change with no clinical significance. Whether a longer period of daycare service restriction is necessary to affect cognition and daily activity function in elderly people with dementia during the COVID-19 pandemic requires further study.

The results of this research have major implications for government departments, social care services, and community-based support initiatives in planning how best to support the population during future pandemics, particularly vulnerable groups, such as those with more impaired physical function at baseline. For example, in elderly people with dementia, efforts should be more focused on maintaining continuity in physical and nutrition service provision in a noncontact format to prevent the worsening of frailty status. Furthermore, while restriction of daycare services is aimed at protecting the most vulnerable, the consequent physical and social inactivity may result in immediate and lasting effects on older adults’ health. Therefore, it is important to consider what factors may mitigate the adverse effects of lockdown on older adults and those living with long-term conditions, such as coping resources and health behaviors, including physical and social engagement, which have been identified as candidate protective factors. However, our study was limited in that protective factors were not examined and, hence, further study is necessary.

This study has some limitations that need to be addressed. First, the sample size was small with a short follow-up period and the study was conducted at a single site, which does not allow generalization of our results. Moreover, there was no control group, so it was not possible to determine whether the change of functional capacity was a result of DCS. Second, in spite of the interruption of DCS, it was not known whether some of the elderly people still had access to other resources, such as contacts through telephone and video consultations during the discontinuation period, which might have helped to preserve their health status. Third, in the current study, we did not assess the burden of neuropsychiatric symptoms. Furthermore, our results were also limited by the single measure that was used to evaluate cognitive decline. Finally, other important parameters, such as the use of drugs and laboratory variables, which may have influenced the physical and mental functions, as well as nutritional deterioration, during the period of confinement, were not included in the analyses. Overall, further larger, prospective studies with a higher number of participants and the inclusion of other subpopulations are needed to clarify the impacts of the COVID-19 lockdown on the functional status of vulnerable older patients and to explore methods that may mitigate the deleterious effects of mandated confinement.

## 5. Conclusions

Our study presented that the objective evidence of physical function by TUG and nutritional status by MNA-SF in older people with dementia were negatively affected by the interruption of daycare services. However, they could still be recovered after resumption of regular program activities within a short-term restriction. We recommend conducting scheduled assessments and interventions to maintain older adults’ daily activities at home to prevent functional decline and malnutrition during any future period of confinement.

## Figures and Tables

**Figure 1 healthcare-10-01744-f001:**
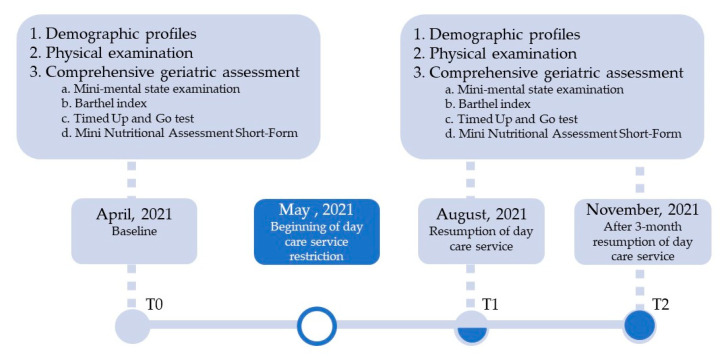
Overview of the study process and data collection periods.

**Table 1 healthcare-10-01744-t001:** Baseline characteristics of the participants.

Demographic Characteristics	Mean ± SEM	Median (IQR)
Age (years)	80.2 ± 2.4	81 (74.5–86)
Sex, n (%)		
Male	7	(41.2%)
Female	10	(58.8%)
Body Mass Index (kg/m^2^)	24.2 ± 0.7	24.3 (22.8–25.3)
Educational level, n (%)		
Illiterate	2	(11.8%)
Literate	15	(88.2%)
Marital Status, n (%)		
Married	6	(35.3%)
Widowed	10	(58.8%)
Divorced	1	(5.9%)
Comorbid disease number, n (%)	3	(2–4)
Comorbidities, n (%)		
Hypertension	8	(47.1%)
Diabetes mellitus	5	(29.4%)
Parkinson’s disease	4	(23.5%)
Cerebrovascular accident	3	(17.6%)
Cardiovascular diseases	2	(11.8%)
Long-term care disability level, n (%)		
2	2	(11.8%)
3	1	(5.9%)
4	8	(47.1%)
5	3	(17.6%)
6	3	(17.6%)
Clinical dementia rating, n (%)		
0.5	6	(35.3%)
1	7	(41.2%)
2	3	(17.6%)
3	1	(5.9%)

**Table 2 healthcare-10-01744-t002:** Change of physical and cognitive functions, and nutritional status at baseline and after restriction and resumption of daycare services.

	Baseline		After 3-Month Restriction		After 3-Month Resumption		*p* Value
	Mean ± SEM	Median (IQR)	Mean ± SEM	Median (IQR)	Mean ± SEM	Median (IQR)	
Body mass index	24.2 ± 0.7	24.3 (22.8–25.3)	23.6 ± 0.7	23.7 (22.1–25.3)	23.9 ± 0.7	24.2 (22.6–25.1)	0.215
Mini-mental state examination	21.9 ± 1.4	21.0 (18.5–28.5)	20.8 ± 1.5	19.0 (17.0–27.5)	20.9 ± 1.4	20.0 (17.0–27.0)	0.019
Five-item geriatric depression scale	0.1 ± 0.1	0 (0–0)	0.4 ± 0.2	0 (0–0.5)	0.4 ± 0.1	0 (0–1.0)	0.186
Barthel index of activities of daily living	83.5 ± 3.1	85.0 (72.5–95.0)	82.6 ± 3.3	85.0 (70.0–95.0)	81.5 ± 3.3	85.0 (70.0–92.5)	0.021
Lawton instrumental activities of daily living scale	2.8 ± 0.4	3.0 (1.5–5.0)	2.8 ± 0.5	2.0 (1.5–5.0)	2.7 ± 0.5	2.0 (1.0–5.0)	0.472
6 m walking speed	0.8 ± 0.1	0.9 (0.6–0.9)	0.8 ± 0.1	0.9 (0.5–1.1)	0.8 ± 0.1	0.9 (0.6–1.0)	0.133
Timed up-and-go test	25.0 ± 9.2	12.8 (10.5–19.4)	31.0 ± 13.4	16.3 (11.8–21.0)	26.1 ± 9.6	13.4 (11.5–19.5)	0.047
Functional reach test	22.9 ± 1.7	22.3 (17.9–26.8)	20.8 ± 1.7	19.4 (15.7–24.6)	22.3 ± 1.7	20.9 (18.4–24.8)	0.368
Mini nutritional assessment short form	12.5 ± 0.3	13.0 (12.0–13.0)	11.4 ± 0.4	12.0 (10.5–12.0)	12.4 ± 0.3	13.0 (11.5–13.0)	0.001
EQ-5D utility index	0.805 ± 0.051	0.877 (0.625–1)	0.798 ± 0.052	0.756 (0.61–1)	0.817 ± 0.046	0.833 (0.625–1)	0.268
EQ-visual analogue scale	83.9 ± 3.1	85.0 (75–95)	82.5 ± 3.6	85.0 (77.5–95)	86.4 ± 3.4	90.0 (77.5–99.0)	0.463

**Table 3 healthcare-10-01744-t003:** The correlation between baseline parameters of CGA and physical and cognitive function after 3 months of daycare services restriction and resumption.

	After 3-Month Restriction				After 3-Month Resumption			
	Timed Up-and-Go Test		Mini Nutritional Assessment Short-Form		Timed Up-and-Go test		Mini Nutritional Assessment Short-Form	
	r_s_	*p* Value	r_s_	*p* Value	r_s_	*p* Value	r_s_	*p* Value
Baseline parameters of CGA								
Body mass index	−0.10	0.701	0.30	0.243	−0.01	0.978	0.73	0.001
Mini-mental state examination	−0.15	0.565	0.09	0.745	−0.28	0.271	0.18	0.484
Five-item geriatric depression scale	−0.10	0.697	−0.32	0.205	−0.31	0.232	−0.46	0.065
Barthel index of activities of daily living	−0.61	0.009	0.26	0.321	−0.66	0.004	0.48	0.051
Lawton instrumental activities of daily living scale	−0.54	0.025	0.01	0.955	−0.75	0.001	0.13	0.615
6 m walking speed	−0.84	<0.001	−0.22	0.386	−0.70	0.002	−0.05	0.834
Functional reach test	−0.40	0.124	0.49	0.056	−0.48	0.057	0.02	0.943
EQ-5D utility index	−0.60	0.011	0.26	0.319	−0.55	0.023	0.24	0.349
EQ-visual analogue scale	−0.74	0.001	0.08	0.761	−0.63	0.007	−0.20	0.436

## Data Availability

The datasets used and analyzed during the current study are available from the corresponding authors on reasonable request.

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
