# Peer review of "Impact of Daycare Service Interruption during COVID-19 Pandemic on Physical and Mental Functions and Nutrition in Older People with Dementia"

_healthcare, 2022, doi:10.3390/healthcare10091744_

Round 1
Reviewer 1 Report
Dear Authors
Abstract: you could mention here how many patients are in the homecare so that we establish if the sample of 17 is sufficient from the word go without having to peruse the methods section
Introduction: Just as a way to signpost your study, at the end of the introduction section have a clear aim set out before you can now focus on the methods section. It can be last paragraph/sentence for the introduction part
Methods: In section 2.4 is measures. You have mentioned them but has not provided their reliability. Please mention their reliability in past studies to affirm they are quality and can be relied upon in this study
Author Response
Point 1: Abstract: you could mention here how many patients are in the homecare so that we establish if the sample of 17 is sufficient from the word go without having to peruse the methods section
Response 1: Thank you for pointing this out. The revised description has been
corrected in the abstract section of revised manuscript with tracked changes at page 1, line 19 to 20.
Point 2: Introduction: Just as a way to signpost your study, at the end of the introduction section have a clear aim set out before you can now focus on the methods section. It can be last paragraph/sentence for the introduction part
Response 2: Thank for your suggestion. We have revised the content in the introduction section of revised manuscript with tracked changes at page 2, line 86 to 89.
Point 3: Methods: In section 2.4 is measures. You have mentioned them but has not provided their reliability. Please mention their reliability in past studies to affirm they are quality and can be relied upon in this study
Response 3: Thank you for pointing this out. Please see in the methods section of revised manuscript with tracked changes at page 4, line 155 to 157.

Reviewer 2 Report
The analyzed topic is very interesting, important and on time.
I believe that the research conducted is extremely important and I'm glad, that the Authors want to share it.
The title is adequate to the research problem being undertaken. The technical part of the article does not raise any objections. The work is aesthetic. The correct terminology and key words was used.Footnotes and bibliography are correctly formulated.
The article has been correctly divided into relevant sections, and their content coincides with their titles.
The authors focused very much on the research part, but devoted far too little attention to the theoretical part.
I propose to broaden the Background part significantly and to separate Literature Review in it.
In my opinion, the article uses a unsatisfactory number of references from international literature (33 items only). This is definitely not enough. This par should be supplemented.
The literature references used are current and relevant with the topic.
There is no broader description of the health and demographic situation of the Taiwan people and of the health care system. Especially in the context of readers who have no knowledge in this area. The authors also did not sufficiently substantiate why this topic is important.
In the methodological part, the calculation formulas used in the analysis are not presented.
The statements of 17 respondents were analyzed. However, we do not know what the percentage of the population that could be surveyed is, and whether these data are statistically significant or not.
I believe that the analysis of the tables in the Results section is too poor. The test results were not described in detail. This needs to be completed.
In my opinion, the Conclusion part does not exist. This two sentences this is not a conclusion.
The purpose of the paper is defined at the beginning of the paper, but not addressed in the last part.
The hypothesis were not described. It is hard to refer to an article when one does not know the purpose and assumed research hypothesis.
Author Response
Point 1: The authors focused very much on the research part, but devoted far too little attention to the theoretical part. I propose to broaden the Background part significantly and to separate Literature Review in it.
Response 1: Thank for your suggestion. We have revised the content in the introduction section of revised manuscript with tracked changes at page 1, line 31 to page 2, line 46.
Point 2: In my opinion, the article uses a unsatisfactory number of references from international literature (33 items only). This is definitely not enough. This par should be supplemented. The literature references used are current and relevant with the topic.
Response 2: Thank you for pointing this out. The revised description has been corrected in the revised manuscript. Please see the references section at page 9, line 336 to page 11, line 447 (total 42 items).
Point 3: There is no broader description of the health and demographic situation of the Taiwan people and of the health care system. Especially in the context of readers who have no knowledge in this area. The authors also did not sufficiently substantiate why this topic is important.
Response 3: Thank for your suggestion. We have revised the content in the introduction section of revised manuscript with tracked changes at page 1, line 36 to 42.
Point 4: In the methodological part, the calculation formulas used in the analysis are not presented. The statements of 17 respondents were analyzed. However, we do not know what the percentage of the population that could be surveyed is, and whether these data are statistically significant or not. I believe that the analysis of the tables in the Results section is too poor. The test results were not described in detail. This needs to be completed.
Response 4: Thank you for pointing this out. We have revised the content in the methods section of revised manuscript with tracked changes at page 3, line 101 to 102, line 110 to 112, line 155 to 157. The results and tables were also revised. Please see in the results section of revised manuscript with tracked changes at page 6, line 187 to 191, line 200 to 202 and table 3.
Point 5: In my opinion, the Conclusion part does not exist. This two sentences this is not a conclusion. The purpose of the paper is defined at the beginning of the paper, but not addressed in the last part. The hypothesis were not described. It is hard to refer to an article when one does not know the purpose and assumed research hypothesis.
Response 5: Thank for your suggestion. The revised description has been presented in the conclusion section of revised manuscript with tracked changes at page 9, line 305 to 307.

Round 2
Reviewer 2 Report
I don't believe the authors have made significant corrections to their paper. Several publications have been added (particularly 9 items), but in my opinion this is still not a fair literature review. The corrections made are symbolic (for example subtitles added). There was no in-depth improvement of the text. Therefore, I'm listing again what needs to be corrected:- far too little attention to the theoretical part.
I propose to broaden the Background part significantly and to separate Literature Review in it.
In my opinion, the article uses a unsatisfactory number of references from international literature (now 42 items only). This is definitely not enough. This part should be supplemented.
- the authors incorrectly added new literature without changing the numbering. We have footnote 21, 23, 26 and then the first time is footnote 22, 24, 25, etc.
- There is no broader description of the health and demographic situation of the Taiwan people and of the health care system. Especially in the context of readers who have no knowledge in this area. The authors also did not sufficiently substantiate why this topic is important.
- In the methodological part, the calculation formulas used in the analysis are not presented.
The statements of 17 respondents were analyzed. We don't know if the data are statistically significant or not.
- I believe that the analysis of the tables content in the Results section is too poor. The test results were not described in detail. This needs to be completed.
- In my opinion, the Conclusion part still does not exist. Maybe better option will be combining Discussion and Conclusion part into one section?
- I still don't know if the goal set at work has been achieved.
The Authors included a research hypothesis, but did not refer to it, whether they were able to confirm it or not.
Author Response
Point 1: - far too little attention to the theoretical part. I propose to broaden the Background part significantly and to separate Literature Review in it. In my opinion, the article uses a unsatisfactory number of references from international literature (now 42 items only). This is definitely not enough. This part should be supplemented.
Response 1: Thank for your suggestion. We have revised the content in the introduction section of revised manuscript with tracked changes at page 1, line 31 to page 2, line 61. Besides, the number of references were also added to 51 items.
Point 2:- the authors incorrectly added new literature without changing the numbering. We have footnote 21, 23, 26 and then the first time is footnote 22, 24, 25, etc.
Response 2: Thank you for pointing this out. The revised description has been corrected in the revised manuscript.
Point 3: - There is no broader description of the health and demographic situation of the Taiwan people and of the health care system. Especially in the context of readers who have no knowledge in this area. The authors also did not sufficiently substantiate why this topic is important.
Response 3: Thank for your suggestion. We have revised the content in the introduction section of revised manuscript with tracked changes at page 1, line 39 to page 2, line 61.
Point 4: - In the methodological part, the calculation formulas used in the analysis are not presented. The statements of 17 respondents were analyzed. We don't know if the data are statistically significant or not.
Response 4: Thank for your suggestion. We have revised the content in the methodological section of revised manuscript with tracked changes at page 4, line 145 to page 5, line 188.
Point 5: - I believe that the analysis of the tables content in the Results section is too poor. The test results were not described in detail. This needs to be completed.
Response 5: Thank for your suggestion. We have revised the content in the results section of revised manuscript with tracked changes at page 4 to page7. Moreover, we revised the table content.
Point 6: - In my opinion, the Conclusion part still does not exist. Maybe better option will be combining Discussion and Conclusion part into one section?
Response 6: Thank you for your suggestion. We have revised the conclusion part of revised manuscript with tracked changes at page 9, line 338 to 355.
Point 7: - I still don't know if the goal set at work has been achieved. The Authors included a research hypothesis, but did not refer to it, whether they were able to confirm it or not.
Response 7: Thank you for your suggestion. We have revised the conclusion part of revised manuscript with tracked changes at page 9, line 338 to 355.
